# Methylation and Expression of *FTO* and *PLAG1* Genes in Childhood Obesity: Insight into Anthropometric Parameters and Glucose–Lipid Metabolism

**DOI:** 10.3390/nu13051683

**Published:** 2021-05-15

**Authors:** Wojciech Czogała, Małgorzata Czogała, Wojciech Strojny, Gracjan Wątor, Paweł Wołkow, Małgorzata Wójcik, Mirosław Bik Multanowski, Przemysław Tomasik, Andrzej Wędrychowicz, Wojciech Kowalczyk, Karol Miklusiak, Agnieszka Łazarczyk, Przemysław Hałubiec, Szymon Skoczeń

**Affiliations:** 1Department of Pediatric Oncology and Hematology, University Children’s Hospital of Krakow, 30-663 Krakow, Poland; czogala@tlen.pl (W.C.); malgorzata.czogala@uj.edu.pl (M.C.); wojciech.strojny@mp.pl (W.S.); 2Department of Pediatric Oncology and Hematology, Faculty of Medicine, Jagiellonian University Medical College, 30-663 Krakow, Poland; 3Center for Medical Genomics—OMICRON, Jagiellonian University Medical College, 30-663 Krakow, Poland; gracjan.wator@uj.edu.pl (G.W.); pawel.wolkow@uj.edu.pl (P.W.); 4Department of Pediatric and Adolescent Endocrinology, Faculty of Medicine, Jagiellonian University Medical College, 30-663 Krakow, Poland; malgorzata.wojcik@uj.edu.pl; 5Department of Medical Genetics, Faculty of Medicine, Jagiellonian University Medical College, 30-663 Krakow, Poland; miroslaw.bik-multanowski@uj.edu.pl; 6Department of Clinical Biochemistry, Faculty of Medicine, Jagiellonian University Medical College, 30-663 Krakow, Poland; p.tomasik@uj.edu.pl; 7Department of Pediatrics, Gastroenterology and Nutrition, Faculty of Medicine, Jagiellonian University Medical College, 30-663 Krakow, Poland; andrzej.wedrychowicz@uj.edu.pl; 8Student Scientific Group of Pediatric Oncology and Hematology, Jagiellonian University Medical College, 30-663 Krakow, Poland; w.kowalczyk@student.uj.edu.pl (W.K.); karolmiklusiak@gmail.com (K.M.); agnieszka.lazarczyk@student.uj.edu.pl (A.Ł.); przemyslawhalubiec@gmail.com (P.H.)

**Keywords:** epigenetics, expression, *FTO*, *PLAG1*, obesity, insulin resistance, children

## Abstract

The occurrence of childhood obesity is influenced by both genetic and epigenetic factors. *FTO* (*FTO* alpha-ketoglutarate dependent dioxygenase) is a gene of well-established connection with adiposity, while a protooncogene *PLAG1* (*PLAG1* zinc finger) has been only recently linked to this condition. We performed a cross-sectional study on a cohort of 16 obese (aged 6.6–17.7) and 10 healthy (aged 11.4–16.9) children. The aim was to evaluate the relationship between methylation and expression of the aforementioned genes and the presence of obesity as well as alterations in anthropometric measurements (including waist circumference (WC), body fat (BF_kg) and body fat percent (BF_%)), metabolic parameters (lipid profile, blood glucose and insulin levels, presence of insulin resistance) and blood pressure. Expression and methylation were measured in peripheral blood mononuclear cells using a microarray technique and a method based on restriction enzymes, respectively. Multiple regression models were constructed to adjust for the possible influence of age and sex on the investigated associations. We showed significantly increased expression of the *FTO* gene in obese children and in patients with documented insulin resistance. Higher *FTO* expression was also associated with an increase in WC, BF_kg, and BF_% as well as higher fasting concentration of free fatty acids (FFA). *FTO* methylation correlated positively with WC and BF_kg. Increase in *PLAG1* expression was associated with higher BF%. Our results indicate that the *FTO* gene is likely to play an important role in the development of childhood adiposity together with coexisting impairment of glucose-lipid metabolism.

## 1. Introduction

Obesity in children is an increasing civilization problem, which is based on lifestyle, socioeconomic, and genetic factors [1]. Currently, the prevalence of obesity among children aged 2–19 is about 17% and it becomes an important therapeutical challenge [2]. Similarly, obesity among children in Poland is at the level of about 12 percent with an upward trend in recent years [3]. Although the impact of environmental factors, like a sedentary lifestyle and excessive nutrient supply, has a large contribution to the end result, more attention should be paid to other relevant obesity-related factors [4]. Many studies, also in the pediatric population, focus on single nucleotide polymorphisms (SNPs) in genes with a well-described relationship with obesity, such as *FTO* [5,6,7,8,9,10,11,12,13,14,15], or genes, whose role in obesity is recently gradually being discovered, as the *PLAG1* gene [16,17,18]. However, very few studies address the impact of epigenetics, including the DNA methylation of CpG dinucleotides, on this phenomenon [19,20]. In the face of the still growing importance of this problem, the influence of epigenetics on childhood obesity is becoming an increasingly discussed issue [21,22].

Therefore, we focus on the relationship between obesity and the degree of methylation of *FTO* (*FTO* alpha-ketoglutarate dependent dioxygenase) and *PLAG1* (pleomorphic adenoma gene 1 zinc finger) genes. Their methylation has a significant impact on the activity of the protein products they encode. The *FTO* gene has a well-described relationship with obesity. In our study, we focused on the methylation analysis of region 1 (*FTO* upstream) 16:53703684–53703899 and region 3 54054572 (CG). The *PLAG1* gene is important in modulating the activity of metabolic processes, so it may have a potential impact on the occurrence of obesity in the pediatric population, which has been recently observed [23]. In our study, we examined the exon 1 of the *PLAG1* gene, region 8:56211059–56211208. Genome reference sequence used was GRCh38.p13.

As both genes influence the metabolism of fat tissue cells and are strictly related to body weight, we decided to study their regulation in pediatric obesity. Therefore, the aim of our study was to analyze the genome expression and methylation of *FTO* and *PLAG1* genes in children with obesity. Additionally, we investigated the effect of *FTO* gene methylation and both gene expression on patients’ anthropometric parameters, such as BMI or waist circumference, and biochemical parameters, e.g., cholesterol level, triglycerides, free fatty acids, and glucose level in the oral glucose tolerance test.

## 2. Materials and Methods

Sixteen obese children (6 boys, 10 girls) aged 6.6–17.7 (median 14.6) years, and 10 normal-weight preadolescent children (4 boys, 6 girls) aged 11.4–16.9 (median 14.2) years (Appendix A), were included in the analysis. The diagnosis of obesity was made in the Department of Children and Adolescent Endocrinology, Institute of Pediatrics, Jagiellonian University Medical College. Obesity was defined as a BMI at or above the 95th percentile for children and teens of the same age and sex. BMI was calculated by dividing a person’s weight in kilograms by the square of height in meters. The inclusion criteria were obesity developed before puberty and medical history negative for acute or chronic disease excluding obesity. Exclusion criteria were secondary obesity because of a single gene mutation, endocrine disorders, other systemic diseases, or pharmacotherapy. Mean BMI percentile in this group was 99.78 ± 0.41.

The control group was recruited from children of medical staff and families of the patients. All children had a medical history negative for chronic or acute diseases, including obesity. Mean BMI percentile in this group was 54.77 ± 33.45.

The post-hoc power calculation from the critical Z value (for the differences of means for *FTO* gene expression) showed a power of 0.95 to detect the differences in the means between groups. For *PLAG1* expression the calculated power was 0.86 and for *FTO* gene region 1 methylation power was 0.69.

### 2.1. Microarray Analysis

Blood samples (1.5 mL) were collected from each child from the obesity and control groups. Total RNA extraction from blood mononuclears was performed using RiboPure Blood Kit (Ambion, Life Technologies, Carlsbad, CA, USA). Whole genome expression was assessed using GeneChip Human Gene 1.0 ST Arrays (Affymetrix, Santa Clara, CA, USA). All procedures were performed according to the manufacturer’s protocol (GeneChip Whole Transcript sense Target Labeling Assay Manual, Version 4). The selection of *FTO* and *PLAG1* genes for further analysis was based on the fact that initially the primary goal of the study was to assess gene methylation in obese children and, as we did not possess necessary resources to carry out an epigenome-wide association study, two genes were chosen: one of well-established (*FTO*), and one of probable (*PLAG1*) association with obesity. Additionally, as we assumed that it would be reasonable to study gene methylation together with expression, and we extracted the expression data from microarray analysis which we carried out initially for the purpose of another research. Genome-wide transcriptome analysis has been already performed in the pediatric cohort, as reported by Keustermans et al. [24].

### 2.2. Methylation Analysis

#### 2.2.1. DNA and Selection of Fragments

Genomic DNA was isolated from peripheral blood using MasterPure DNA Purification Kit for Blood (Epicentre). *FTO* (Chr16) methylation of region 1 [upstream, 53703684–53703899] and region 3 [54054572 (CG)] was tested with the method based on methylation-dependent (MDRE) and methylation sensitive (MSRE) restriction enzymes. MSREs are more sensitive to low levels of methylation in contrast to MDREs, which are more capable of detecting high levels of methylation [25,26]. Target regions of both genes were chosen based on methyl-DIP data available in Ensembl database [27]. We picked up sequences that contain CCGG palindrome targets for MDRE and MSRE enzymes. Sequences of primers used for qPCR are shown in Table 1. Primers were designed using PrimerQuest (Integrated DNA Technologies) as well as Primer3 (Whitehead Insitute for Biomedical Research)—[28]. Additionally, methylation of *PLAG1* gene, region 1 [(*PLAG1* exon1) 56211059–56211208] was investigated. *PLAG1* methylation served as an additional control for *FTO*.

#### 2.2.2. Restriction Enzyme Digestion

Briefly, for MSRE reaction we used HpaII enzyme in combination with MspI (EpiJET, Thermo, Waltham, MA, USA). Both enzymes were isoschizomers which recognize CCGG sites. Methylation of the internal CpG site blocks a cleavage with HpaII, whereas cleavage with MspI is unaffected. Digestions with both enzymes were carried out overnight at 37 °C and terminated at 90 °C for 10 min. Digestion efficiency were controlled using unmethylated and methylated pUC19/SmaI DNA. We used 50 ng and 100 ng of genomic DNA to perform digestion with FspEI (MDRE) and HpaII (MSRE) enzymes, respectively. For MDRE, we used FspEI modification-dependent endonuclease (New England BioLabs, Ipswich, MA, USA). The reaction was carried out with 30 µL in the presence of 2.5 U of enzyme and reaction buffer supplemented with Enzyme Activator Solution and BSA. After 4 h of incubation at 37 °C the reaction was stopped for 20 min at 80 °C.

#### 2.2.3. qPCR

Real-time experiments were performed according to MIQE guidelines. For all reactions, we applied intercalating dye chemistry with SYBR Green as well as hot start iTaq DNA polymerase (iTaq Universal SYBR Green Supermix, Bio-Rad, Hercules, CA, USA). All reactions were run in triplicate on CFX384 Touch Real-Time PCR Detection System (Bio-Rad). On each reaction plate, serial dilutions of control DNA were run to establish PCR efficiency and correct calculations.

### 2.3. Methylation Data Processing and Statistical Analysis

All downstream data processing and statistical analyses were performed with the statistical software R [29] together with the methylumi and limma [30] packages of the Bioconductor project [31].

Moderated t-statistics for each contrast and probe was created using an empirical Bayes model as implemented in limma (eBayes command). *p*-values were adjusted for multiple comparisons as proposed by Benjamini and Hochberg [32] and an adjusted *p*-value > 0.05 was considered nonsignificant (ns).

### 2.4. Anthropometric Measurements

Body weight and height were measured to the nearest 0.1 kg and 0.1 cm, respectively, using a stadiometer and a balanced scale. Waist circumference was measured under standardized procedures as instructed by the WHO (midpoint between the lower costal margin and iliac crest). All measurements were conducted by an anthropometrist. Online World Health Organization (WHO) BMI calculators were used to determine body mass index (BMI) and BMI percentile (BMI_Perc) [33]. The results were then confronted with regional reference values and the reference values published by WHO [33,34,35]. Bioelectrical impedance analysis (BIA) was used to measure the body fat mass (BF_kg) and body fat percentage (BF_%). The method described by Kushner and Schoeller [36] was then applied to calculate the values of the parameters.

### 2.5. Study Protocol

Blood concentrations of glucose, free fatty acids, and insulin were measured at fasting as well as at 60- and 120-min of the standard oral glucose tolerance test (OGTT) performed using 1.75 g of anhydrous glucose per kilogram of body weight (maximum of 75 g). Plasma concentrations of total cholesterol, low-density lipoprotein (LDL), high-density lipoprotein (HDL), and triglycerides were measured only at fasting. EDTA-and-aprotinin-containing tubes (BectonDickinson, Plymouth, UK) as well as tubes with no anticoagulant were used to collect blood samples. After an immediate delivery to the laboratory, the tubes were centrifuged for 15 min at 3000 rpm using a horizontal rotor.

### 2.6. Laboratory Measurements

Vitros 5.1 dry chemistry analyzer (Johnson & Johnson, Ortho-Clinical Diagnostics, Inc., Rochester, NY, USA) was used to evaluate the fasting blood glucose as well as blood concentrations of triglycerides, free fatty acids, total cholesterol, HDL, and LDL. Insulin levels were measured using immunoradiometric kits (BioSource, Nivelles, Belgium). Prior to the analysis, samples of insulin were kept at −80 °C. 

### 2.7. HOMA-IR

The insulin resistance homeostasis model assessment (HOMA-IR) index was calculated using a standard (glucose [mmol/L] × insulin [µU/mL])/22.5 equation. A value > 2.5 of HOMA-IR index was considered as indicative of insulin resistance.

### 2.8. Blood Pressure

Systolic blood pressure (SBP) and diastolic blood pressure (DBP) were measured three times in 3-min intervals using an electronic sphygmomanometer and mean values were calculated. BP values were assessed using centile charts according to “Age-based Pediatric Blood Pressure Reference Charts” Retrieved 26 February 2021 from the Baylor College of Medicine, Children’s Nutrition Research Center, Body Composition Laboratory WebSite [37].

### 2.9. Statistical Analysis

Continuous biochemical and clinical variables are presented as mean values and standard deviation or as median values and quartiles, as appropriate. Categorical variables are presented as frequencies and percentages. Shapiro–Wilk test was used to assess the normal distribution of continuous variables. To examine the differences between independent groups the analysis of covariance (ANCOVA) was used. To assess the correlations between two continuous variables, Spearman rank correlation coefficient was calculated. To adjust for possible influence of children’s age and sex on the calculated relationships, multiple regression models were constructed (logistic regression for dichotomous variables and linear or linearized non-linear regression for continuous variables, as appropriate). Two-sided *p* values < 0.05 were considered statistically significant. 

Gene expression data were RobustMulti-array Average (RMA)-normalized and presented as mean and standard deviation. ANOVA was used to examine the differences in gene expression between two independent groups. Benjamini-Hochberg (B-H)-corrected *p* values < 0.05 were considered statistically significant. Statistical analysis was performed using Statistica 13.3 (Statsoft Inc., Tulsa, OK, USA).

The Permanent Ethical Committee for Clinical Studies of the Medical College of the Jagiellonian University approved the study protocol. All parents, adolescent patients, and adult patients signed a written informed consent before blood sample collection. The study conforms with The Code of Ethics of the World Medical Association (Declaration of Helsinki), printed in the British Medical Journal (18 July 1964).

## 3. Results

Characteristics of the study groups are presented in Table 2. Statistically significant differences in mean values between the groups were noticed in terms of both BMI (32.74 ± 4.59 in the obesity group and 20.08 ± 3.3 in the control group, *p* < 0.001) and BMI percentile (99.78 ± 0.41 vs. 54.77 ± 33.45, *p* = 0.002) based on which patients were assigned to groups. The next parameters differentiating the group were waist circumference (97.46 ± 14.23 vs. 72.18 ± 9.24, *p* < 0.001), percentile of waist circumference (94.77 ± 6.25 vs. 46 ± 33.9, *p* = 0.001), BF_kg (37.89 ± 11.28 vs. 15.23 ± 9.95, *p* < 0.001), and BF_% (40.72 ± 8.25 vs. 23.96 ± 11.65, *p* = 0.008). In terms of the lipid profile, a statistically significant difference was found for triglycerides (1.68 ± 0.52 vs. 0.9 ± 0.28, *p* < 0.001). The insulin concentrations showed significant difference between the groups in all three measurements taken during the OGTT, which also entailed a difference in the value of the HOMA-IR index (6.23 ± 3.25 vs. 2.43 ± 1.19, *p* = 0.001). Differences between the groups are also observed in the case of mean systolic blood pressure (mmHg) (118 ± 9.23 vs. 108.9 ± 5.43, *p* = 0.006) and its percentile (73.14 ± 8.195 vs. 44.7 ± 16.97, *p* = 0.001).

### 3.1. Genes Expression

A comparison of the obesity and control groups revealed higher mean expression of *FTO* gene in obesity (317.37 ± 1.32 vs. 200.85 ± 1.4, *FC* = 1.58, *p*(B-H) = 0.008). 

The analysis of the expression of *PLAG1* gene also showed a significantly higher expression in the obesity group compared to the control group (89.88 ± 1.32 vs. 65.8 ± 1.17, *FC* = 1.37, *p*(B-H) = 0.036). 

Correlations of *FTO* and *PLAG1* expression levels with the studied parameters are presented in Table 3, Table 4 and Table 5.

### 3.2. Methylation Results

The mean and median methylation values for *FTO* region 1 in the obese and control groups were 5.57%/4.53% and 2.4%/1.83% (*p* = 0.041), respectively. For region 3, the results were: 44.17%/45.84% and 47.09%/45.39% (*p* = 0.256), respectively. Spearman’s correlations of *FTO* region 1 methylation results with the studied parameters are presented in Table 3, Table 4 and Table 5 and Figure 1. 

The mean and median methylation percentage for *PLAG1* region 1 in obese and control groups were 0.09%/0.09% and 0.08%/0.05% (*p* = 0.48), respectively. Due to the lack of a statistically significant difference, Spearman’s correlations with studied parameters were not calculated. 

### 3.3. Spearman’s Correlation Results of Methylation Level and Expression of FTO Gene and PLAG1 Gene

In the case of the *FTO* gene, Spearman’s correlation coefficient r between the methylation level of region 1 with gene’s expression is *r* = 0.381, with a borderline significance *p* = 0.055. 

For *FTO* region 3 methylation level, Spearman’s correlation coefficient r with expression level is *r* = 0.635 (*p* = 0.006), however, for this region there was a large loss of data due to low digestion rate of the samples (we only have this data for 16 patients, 65.38%). For this reason, in the further part of the analysis, we did not compare the degree of region 3 methylation with the anthropometric and biochemical parameters.

The correlation between the methylation level of the tested *PLAG1* gene fragment and the level of its expression was not found *(r* = −0.067, *p* = 0.797), which suggests the lack of influence of this gene fragment methylation on the amount of the synthesized mRNA.

### 3.4. Spearman’s Correlation Results of Anthropometric Parameters with FTO Gene Methylation, Expression and PLAG1 Gene Expression

In the case of the *FTO* gene region 1 methylation level, Spearman’s correlation coefficient r with statistical significance was found for the BMI percentile (*r* = 0.428, *p* = 0.029) and waist circumference percentile (*r* = 0.504, *p* = 0.014) (Table 3, Figure 1).

The contrary results were found for the expression of the *FTO* gene. A statistically significant positive correlation was found for all tested parameters, which is presented in Table 3. It is particularly remarkable in the case of BF_kg (*r* = 0.704, *p* < 0.001) and BF_% (*r* = 0.837, *p* < 0.001), which confirms the *FTO* expression is associated with a predisposition to obesity (Table 3, Figure 2).

In the case of *PLAG1* expression, statical significance was shown for BF_% (*r* = 0.523, *p* = 0.022), BMI (*r* = 0.445, *p* = 0.023), and BMI SD (*r* = 0.430, *p* = 0.028) (Table 3, Figure 3).

Correlations of *FTO* gene region 1 methylation, *FTO* expression, and *PLAG1* expression with the mean systolic blood pressure, mean diastolic blood pressure, and their percentiles; however, the results were not statistically significant (Appendix A).

### 3.5. Spearman’s Correlation Results of Biochemical Parameters with FTO Gene Region 1 Methylation, FTO Expression, and PLAG1 Gene Expression

Correlations of the *FTO* gene region 1 methylation, *FTO* expression, and *PLAG1* gene expression with the lipid profile, glucose, and insulin concentrations at the three OGTT points showed a positive correlation coefficient r of *FTO* gene methylation with triglyceride concentrations (*r* = 0.474, *p* = 0.019) and fasting insulin levels (*r* = 0.428, *p* = 0.037)-(Table 4, Figure 1).

Similar results were found for the expression of *FTO* gene and fasting free fatty acids levels (*r* = 0.661, *p* = 0.004) and for insulin at 0 min, 60 min, and 120 min of the OGTT test (*r* = 0.566, *p* = 0.004; r = 0.444, *p* = 0.03; *r* = 0.458, *p* = 0.028; respectively). The value of the HOMA-IR index showed a negative correlation with fasting serum glucose level (*r* = −0.566, *p* = 0.004) (Table 4, Figure 2). Correlation with other examined parameters (total cholesterol, LDL, HDL, free fatty acids OGTT 60 min and 120 min, glucose-OGTT 60 min and 120 min) did not reach statistical significance (Appendix A).

The expression of the *PLAG1* gene correlated with the fasting level of free fatty acids (*r* = 0.563, *p* = 0.019) (Table 4, Figure 3).

### 3.6. Effect of FTO Gene Region 1 Methylation, FTO Expression, and PLAG1 Expression on the Presence of Insulin Resistance

Insulin resistance was associated with the expression level of the *FTO* gene (*p*_adj_ = 0.037) (Table 5). No correlation between the *PLAG1* gene expression and the concentration of insulin and glucose was found.

### 3.7. Multiple Regression Analysis

By the multiple regression the relationship between *FTO* gene region 1 methylation and the expression of genes was disclosed. Each model was adjusted for age and sex of children and all interactions between variables were included, if existed.

According to the multiple logistic regression, the *FTO* gene expression was independently associated with obesity and insulin resistance (Table 6).

In the multiple linear (or linearized non-linear) regression most of the associations observed in the correlation analysis were replicated, particularly in case of the *FTO* gene expression (Table 7).

## 4. Discussion

In our study, we found that the expression and methylation of the *FTO* and *PLAG1* genes differ between healthy and obese pediatric patients, however the construction of multiple logistic regression model, aimed to adjust for possible influence of age and sex, confirmed these results only for the increased *FTO* expression in the obesity group. *FTO* expression was also significantly higher in patients with documented insulin resistance. Several anthropometric and metabolic parameters were shown to correlate with *FTO* methylation and the expression of both investigated genes. Multiple linear regression analysis revealed that increase in *FTO* expression is independently associated with growing values of BMI (kg/m^2^, percentile and SDS), waist circumference (cm, percentile), BF_kg, BF_%, and FFA levels; *FTO* methylation correlates positively with waist circumference (cm) and BF_kg; whereas higher *PLAG1* expression predicts elevation of BF_% and FFA levels (the latter of borderline significance).

### 4.1. Expression and Methylation of FTO Gene

Based on searches in the GeneCards—The Human Genes Database [38,39], the *FTO* gene is a nuclear protein of the AlkB-related non-hem iron and 2-oxoglutarate-dependent oxygenase superfamily, but its exact physiological function is unknown. The strong association with a role in the nervous and cardiovascular systems was described, as well as with body mass index, obesity risk, and type 2 diabetes (provided by RefSeq, July 2011) [40]. Diseases associated with *FTO* function include growth retardation, developmental delay, facial dysmorphism, coarse facies, and early death. The *FTO* gene-related functions include glucose energy metabolism and DNA damage reversal. It contributes to the regulation of the global metabolic rate, energy expenditure, and energy homeostasis as well as to the regulation of body size and body fat accumulation [21,22,41,42]. The *FTO* gene is involved in the regulation of thermogenesis and the control of adipocyte differentiation into brown or white fat cells [43].

In the present study, the mRNA level of the *FTO* gene was significantly higher in children with obesity than in the control group. This may confirm its importance in the pathogenesis of obesity. Our results are consistent with the results of a study of placental *FTO* gene mRNA. It showed a significantly elevated expression in infants with high (>4000 g) and medium (3500–3999 g) birth weight, compared to children with low birth weight (<3500 g), according to the criteria established by the researchers [44]. In the study by Gardner et al., a link between high levels of *FTO* mRNA and increased food intake in female children was found [19]. Reports describing the pronounced activity of the *FTO* gene in the hypothalamus areas responsible for controlling food intake, explain the mechanism [45].

A significant relationship between infant birth weight and methylation level of the placental *FTO* gene was also found. The degree of methylation of CpG11 sites was significantly reduced in infants with high birth weight (>4000 g) [44]. The study by Gardner et al. showed no association between the level of *FTO* methylation and the risk of obesity, however, it was performed on a narrow age group of 5–6-year-old African-American children [19]. Our research included older children (6.6–17.7, median 14.6), which may suggest that the impact of the *FTO* gene on the risk of obesity increases with age. This supposition was confirmed by a study conducted on a group of children from 5 to 16 years old, showing a positive influence of the rs1421085 *FTO* gene polymorphism on the BMI as the age of the subjects increased [7]. The fact that it was mainly the variability of region 1 of the *FTO* gene that affected body weight was confirmed by other studies [5,46,47,48].

Comparison of the methylation degree of regions 1 and 3 of the *FTO* gene between the group of obese children and the control group in our study revealed a statistically significant difference in the methylation level of region 1 (5.57%/4.53% and 2.4%/1.83%; *p* = 0.041, respectively), but not of region 3 (44.17%/45.84% and 47.09%/45.39%; *p* = 0.256, respectively). 

Region 1 is a part of the *FTO* gene promoter [27,49]. The protein that is the product of this gene has been shown to bind to the promoter and inhibit transcript formation for this protein. This mechanism is based on the principle of negative feedback loop. A properly functioning mechanism allows to control the weight and maintain it at an appropriate level [50]. It is possible that the increased level of methylation of this region prevents *FTO* protein from binding to the promoter. This could have disrupted the negative feedback, leading to increased mRNA expression observed in the study. 

An interesting observation was the relationship between the level of methylation of the *FTO* gene and its polymorphisms in increasing the risk of obesity. Homozygote rs9939609 AA possessed a definitely higher chance to become obese than heterozygotes AT or homozygotes TT. The increase in risk was particularly evident when the AA genotype was accompanied by enhanced methylation of *FTO* gene. In the case of low *FTO* gene methylation, the difference between AA and TT homozygous variants was abolished [51]. Hence it can be concluded that the paramount importance of the methylation level over the studied polymorphism of *FTO* gene, even though the importance of the polymorphism is also meaningful.

In our study, after adjusting the main variable for age and sex of children, and for any interaction existing between covariates, we only obtained a result close to statistical significance for *FTO* gene methylation and obesity. However, some results link the effect of the *FTO* gene methylation with the risk of obesity. Therefore, it is not clear, whether the *FTO* gene methylation in region 1 contributes to increasing the risk of developing obesity and, indirectly, also diseases associated with excessive body weight.

The documented factor triggering hypermethylation of the *FTO* gene is cigarette smoking by the mother during pregnancy, both at the 16 CpG site [52], and 12 CpG [53]. However, smoking by the father and by the examined person himself does not modify the methylation pattern of this gene [52]. Chikky et al. also showed a positive impact of vitamin B12 supplementation by increasing the level of *FTO* methylation [54].

Numerous studies show a correlation between the *FTO* gene and BMI, but most often they are based on single nucleotide polymorphisms (SNP), and not on differences in methylation or expression of *FTO* gene [55,56,57]. Because the association of *FTO* gene polymorphisms with BMI and body composition was described first, a cluster of single nucleotide polymorphisms (SNPs) in the first intron of the *FTO* gene has already been well-known [57]. An example is the study of Qureshi et al., 2017, in which correlation and analysis of variance tests of rs3751812 *FTO* polymorphism with anthropometric measures showed significant association with BMI and weight only, and not with any of height, waist and hip circumference, and waist to hip ratio. In contrast, our study revealed that *FTO* gene methylation is related to body fat mass, and its expression shows a significant relationship to BMI, waist circumference, and body fat mass. However, Khoshi et al. did not find a genotype-related difference between normal and high BMI groups [58].

An interesting observation was made by Nishida et al., who showed that methylation of the *FTO* gene may depend on the physical activity. In obese subjects who conducted additional resistance training during a 6-month weight-loss program intervention, the methylation rate was significantly increased at one CpG site after, and significantly decreased at four CpG sites in the group without special training [56]. The change in methylation was associated with a variation in gene expression level, which may indicate the importance of physical activity in the degree of methylation and expression of the *FTO* gene.

The effect of *FTO* gene on biochemicals parameters was described mostly based on SNPs, too [55,58,59]. While some studies showed no effect of polymorphisms on the lipid profile [58], others described that the SNP was significantly associated with low levels of HDL and high levels of LDL, but all other parameters, i.e., TC, TG, and VLDL, were not affected [55], or significant differences between *FTO* genotype groups appeared only for serum HDL levels [59]. In our study, higher expression of *FTO* gene was associated with a significantly higher concentration of free fatty acids, but the differences in HDL, LDL or triglycerides levels were not significant.

SNPs have been shown to influence glucose and insulin levels during the oral glucose tolerance test (OGTT) [60]. However, the impact of methylation and expression of *FTO* gene region 1 remains undescribed. Our work showed the correlation between higher *FTO* expression and higher insulin levels, but the results are significant in fasting state only.

In the literature, no significant relationship between *FTO* polymorphism and systolic or diastolic blood pressure was observed, which is consistent with our results—methylation and expression of the *FTO* gene region 1 did not significantly affect blood pressure values [61]. Nevertheless, it is worth mentioning that SNPs of *FTO* gene may modify the effect of obesity on high blood pressure, at least in the Chinese child population [62,63].

### 4.2. Expression and Methylation of PLAG1 Gene

*PLAG1* zinc finger (previously known as pleiomorphic adenoma gene 1) [64,65] is a protooncogene which encodes a zinc finger protein with two putative nuclear localization signals (provided by RefSeq, July 2008) [40]. The GeneCards database [38,39] indicates that the *PLAG1* protein functions as a transcription factor responsible for the upregulation and activation of target genes, such as *IGF2*, leading to uncontrolled cell proliferation. Furthermore, *PLAG1* induction is associated with higher expression of other target genes, such as *CRLF1*, *CRABP2*, *CRIP2*, or *PIGF*. The ectopic *PLAG1* gene expression can trigger the development of lipoblastoma and pleomorphic adenomas of the salivary gland, and its overexpression is frequently observed in hepatoblastoma and AML [66,67]. In addition to the well-described role of *PLAG1* as a driver of carcinogenesis, a possible novel aspect of its function, namely affecting the development of obesity, is currently coming to light. GWAS Catalog [68,69], an online base of genome-wide association studies, indicates several surveys which show the link between polymorphisms of the *PLAG1* gene and certain anthropometric parameters. Namely, the rs72656010 variant seems to be associated with birth weight [16] and lean body mass [17], whereas the rs10958476 SNP shows a correlation with BMI-adjusted hip circumference [18]. This data suggest a potential role of *PLAG1* in the determination of body composition and weight, which in turn leads to a conclusion that the development of obesity could be influenced by its activity as well. Supportive of this idea is the fact that the IGF-2 growth factor, which expression is promoted by *PLAG1* transcriptional activity, has a well-established function as one of the key regulators of both pre- and postnatal growth, possibly affecting adipose tissue metabolism and proliferation by binding to insulin receptors [70]. Street et al. reported higher blood IGF-2 levels in obese children compared to normal-weight prepubescents and demonstrated a positive correlation between IGF-2 serum concentrations and standardized BMI values [71]. In the other study, increased IGF-2 cord blood concentration at birth was related to its higher serum levels at age 5, which in turn were positively associated with current fat mass, with only a marginal correlation with fat-free mass [72]. This indicates that IGF-2 could stimulate the growth of adipose tissue more potently than that of other body parts, which further suggests its influence on the development of obesity. *PLAG1* frameshift mutations, resulting in the production of a truncated, nonfunctional protein, have been recently described to be associated with lower IGF-2 expression, thus leading to growth restriction corresponding to the Silver-Russell Syndrome, which further proved the ability of *PLAG1* to determine certain features by altering IGF-2 protein levels [73]. 

Taken together, these data led us to the assumption that elevated *PLAG1* expression could contribute to the development of adiposity in children. Yet, we failed to confirm this hypothesis, as the multiple regression analysis showed that *PLAG1* expression did not differ significantly between obesity and control groups. It did, however, independently correlate with body fat percent in a positive manner, which remains consistent with the aforementioned observation regarding the influence of IGF-2 on adipose tissue content [72]. These results suggest that *PLAG1* expression could be weakly associated with body composition, but not with the development of childhood obesity.

According to EWAS Atlas [74,75] and EWAS Catalog [76,77], online databases of epigenome-wide association studies, differential methylation of several CpG sites within the *PLAG1* gene result in various phenotypic outcomes regarding certain traits. This indicates that epigenetic changes affect the expression and therefore the function of this proto-oncogene. Since hypomethylation is mostly associated with increased transcriptional activity, we hypothesized that the methylation level of *PLAG1* gene would be lower among obese children. This supposition is supported by recent studies, one of which shows that an increased umbilical cord blood methylation degree at the cg21448513 site within the *PLAG1* gene accounts for lower cord serum levels of leptin, a marker of neonatal adiposity [23]. Furthermore, hypomethylation at the cg01994308 site in peripheral white blood cells seems to correlate with increased waist circumference in adult cohort [78]. Cord blood methylation levels at cg21448513 and cg01994308 sites were also shown to be decreased in infants born to overweight or obese women [79] and women with gestational diabetes mellitus [80], respectively. Both of these maternal conditions have a well-defined positive impact on the risk of obesity in the offspring later in life.

CpG islands within the *PLAG1* promoter investigated in our study turned out to be methylated to a very low extent and, accordingly, there was no significant difference in their methylation level between obese and normal-weight patients. Our results suggest that differential methylation of this region has no influence on the development of childhood adiposity and most likely does not contribute to the regulation of *PLAG1* expression whatsoever. This remains in line with the fact that the studied CpG sites had not been previously reported to be associated with any phenotypic changes. Assessment of other CpGs within the *PLAG1* gene or regulatory sequences, especially the aforementioned cg21448513 and cg01994308, could provide an epigenetic background for the enhanced *PLAG1* transcription in the obesity group.

Literature analysis brings to the conclusion that certain metabolic parameters commonly disrupted in obesity could be affected by changes in *PLAG1* expression. In mouse model studies performed by Declercq et al. *PLAG1* overexpression in pancreatic ꞵ-cells led to hyperplasia of the islets resulting in hyperinsulinemia, which in turn caused compensatory hepatic insulin resistance reflected by increased HOMA-IR values. Glucose levels in the studied mice were decreased or remained normal [81,82]. In humans, *PLAG1* expression was found to be increased in blood samples of children with pre-type 1 diabetes [83,84,85]. In this case, the elevated *PLAG1* protein level could perhaps stimulate ꞵ-cells proliferation to compensate for their loss in the autoimmune process. Conversely, human ꞵ-cells transplanted into mice and subjected to hyperglycemic conditions turned out to express lower amounts of *PLAG1* [83,84,86]. Certain target genes upregulated by *PLAG1* are known to affect glucose, lipid, and blood pressure homeostasis mechanisms. IGF-2 is capable of decreasing glycemia, raising the number of lipid droplets and free cholesterol content in murine liver cells as well as upregulating HMGCR (3-hydroxy-3-methylglutaryl-CoA reductase)—the key enzyme of cholesterol biosynthesis pathway [87,88]. Moreover, IGF-2 to IGFBP1 (insulin-like growth factor binding protein 1) plasma concentrations ratio was shown to correlate positively with the insulin levels and HOMA-IR in the children cohort [71]. On the other hand, lower IGF-2 levels were observed in patients suffering from metabolic syndrome compared to healthy subjects [89]. Blood pressure values are also likely to be affected by IGF-2 action. In pregnant women, a positive correlation between plasma IGF-2 concentration and preeclampsia was shown [90]. Furthermore, IGF2 gene polymorphisms were found to be associated with blood pressure values in obese children [91]. *CRABP2*, another *PLAG1* target gene, is known to affect plasma levels of LDL and total cholesterol [92].

Nevertheless, in our study, the *PLAG1* expression was not linked to the presence of insulin resistance nor the values of blood pressure. In case of biochemical parameters, there was a positive correlation regarding the fasting FFA levels. This association was of borderline significance (*p*-Value for R^2^ = 0.050) in the multiple linear regression analysis. The potential link between *PLAG1* expression and FFA levels could be explained by the fact that increased body fat content is commonly associated with elevated FFA concentrations [93], however, a direct link between *PLAG1* activity and FFA production is also possible, as suggested by a study showing that in bovine adipocytes *PLAG1* upregulates *PLIN1* (perilipin 1) which in turn promotes lipolysis [94].

### 4.3. Limitations

The main limitation of the study was the small sample size, which could affect the validity of the results. Particularly, the power of the study to detect the difference in *FTO* methylation was calculated as 0.69, therefore not reaching the commonly used minimum of 0.80. In the future, it would be advisable to carry out similar experiments on larger cohorts to further confirm the obtained results. In addition, certain outcomes, which did not reach the statistical significance cutoff value, are likely to be found significant when investigated with a population of greater size.

In case of methylation of the *PLAG1* gene, we did not perform correlation analysis with anthropometric measurements, parameters of glucose lipid metabolism, and blood pressure values, because of the very low methylation degree of the studied *PLAG1* region.

## 5. Conclusions

Current data from the literature indicated that *FTO* expression could be elevated in children with obesity and our results confirmed this hypothesis. Moreover, the found positive correlation between *FTO* transcript production and body fat content points out that the increase in BMI occurring in the context of *FTO* expression is indeed caused predominantly by the accumulation of adipose tissue and not lean body mass. Furthermore, we demonstrated that increased *FTO* expression is independently associated with the presence of insulin resistance. Cumulatively, this proves that the *FTO* gene could play an important role not only in the development of adiposity itself, but also in the pathogenesis of obesity complications, although further studies, preferably of prospective design, would be needed to confirm these assumptions.

The age-and-sex-adjusted results suggest that methylation of the investigated region of the *FTO* gene does not seem to provide an epigenetic background for the occurrence of childhood obesity.

The outcomes regarding the *PLAG1* gene show that it is unlikely to play a considerable role in the development of adiposity in children, although it could slightly affect the body composition.

## Figures and Tables

**Figure 1 nutrients-13-01683-f001:**
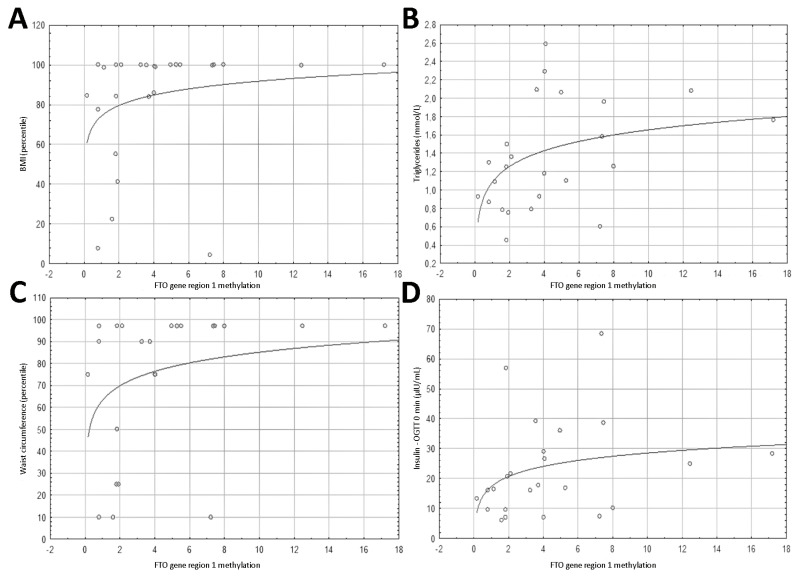
Plots presenting the distribution of the studied parameters depending on the level of *FTO* region 1 methylation [%]. (**A**) BMI percentile; (**B**) triglyceride concentrations [mmol/L]; (**C**) waist circumference percentile; (**D**) insulin—OGTT 0 min (µIU/mL).

**Figure 2 nutrients-13-01683-f002:**
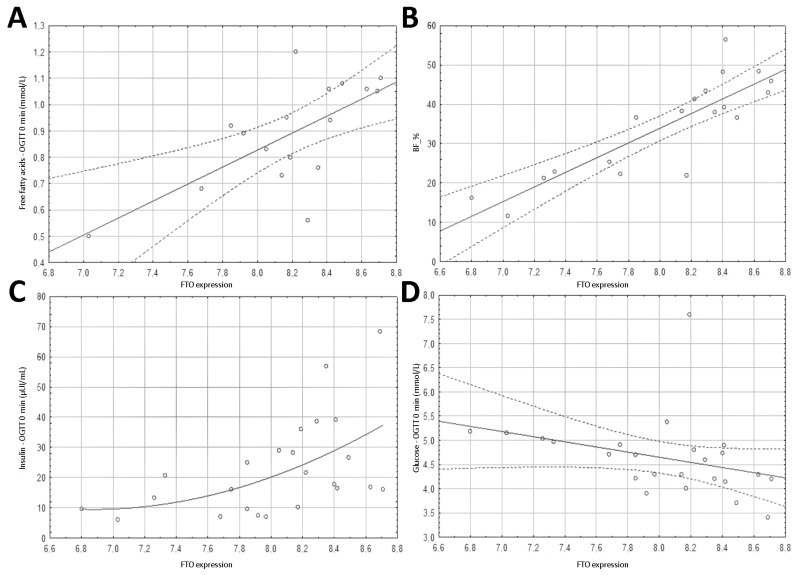
Plots presenting the distribution of data of the studied parameters depending on the level of *FTO* expression (shown as log_2_ of the absolute value of the expression for clarity of the plot). (**A**) Free fatty acids—OGTT 0 min (mmol/L); (**B**) BF_%; (**C**) insulin—OGTT 0 min (µIU/mL); (**D**) glucose—OGTT 0 min (mmol/L).

**Figure 3 nutrients-13-01683-f003:**
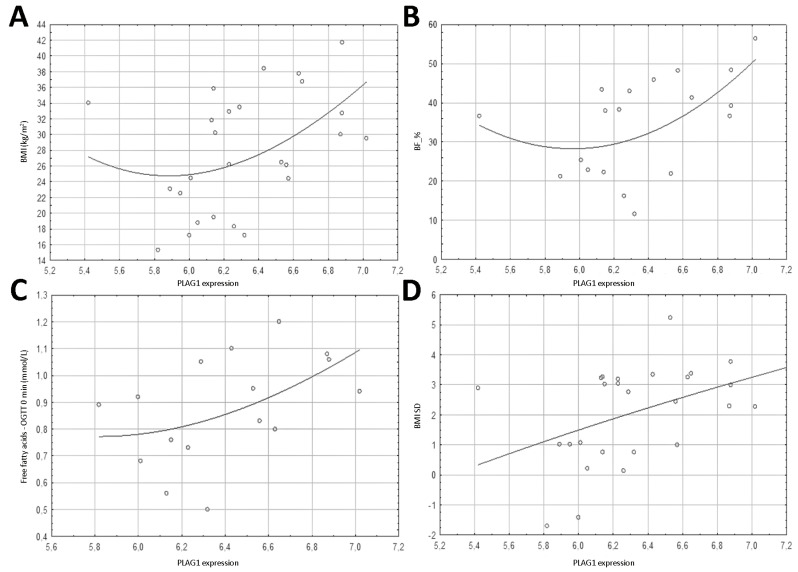
Plots presenting the distribution of data of the studied parameters depending of the level on *PLAG1* expression (shown as log_2_ of the absolute value of the expression for clarity of the plot). (**A**) BMI (kg/m^2^); (**B**) BF_%; (**C**) free fatty acids—OGTT 0 min (mmol/L); (**D**) BMI SD.

**Table 1 nutrients-13-01683-t001:** Sequences of primers used for qPCR reaction.

Gene	Localization	Fragment Size	No. of CCGG Sites	Forward	Reverse
*FTO*	upstream	216 bp	2	CAACTCCAGGGCCTTCTC	GGAGCCTGCCATGTTTCT
*FTO*	3′UTR	214 bp	1	GGAAGGAGAGAATAGAGCCAAG	TCCTCAGCACTTTAGCCTTAC
*PLAG1*	exon1	202 bp	2	ACAATGGCTGCTGGAAAGA	CCCTGATATTTCTCCCGCTAAA

**Table 2 nutrients-13-01683-t002:** Characteristics of the study group.

Baseline Characteristic	Obesity Group *n* = 16	Control Group *n* = 10	*p* Value
boys/girls *n* (%)	6/10 (37.5%/62.5%)	4/6 (40%/60%)	0.609
Age (years)	13.78 ± 2.98	14.32 ± 1.92	0.938
Height (cm)	162.88 ± 15.77	168.01 ± 10.22	0.324
Weight (kg)	88.43 ± 22.48	57.72 ± 15.69	<0.001
BMI (kg/m^2^)	32.74 ± 4.59	20.08 ± 3.3	<0.001
BMI (percentile)	99.78 ± 0.41	54.77 ± 33.45	0.002
Waist circumference (cm)	97.46 ± 14.23	72.18 ± 9.24	<0.001
Waist circumference (percentile)	94.77 ± 6.25	46 ± 33.9	0.001
BMI SD	3.15 ± 0.69	0.27 ± 1.03	<0.001
BF_kg	37.89 ± 11.28	15.23 ± 9.95	<0.001
BF_%	40.72 ± 8.25	23.96 ± 11.65	0.008
Total cholesterol (mmol/L)	4.41 ± 1.04	3.9 ± 0.65	0.159
LDL cholesterol (mmol/L)	2.6 ± 0.92	2.07 ± 0.48	0.084
HDL cholesterol (mmol/L)	1.04 ± 0.23	1.42 ± 0.55	0.138
Triglycerides (mmol/L)	1.68 ± 0.52	0.9 ± 0.28	<0.001
Glucose—OGTT 0 min (mmol/L)	4.59 ± 1	4.71 ± 0.43	0.154
Glucose—OGTT 60 min (mmol/L)	7.37 ± 1.7	6.68 ± 2.05	0.395
Glucose—OGTT 120 min (mmol/L)	6.24 ± 1.67	5.36 ± 1.23	0.152
Insulin—OGTT 0 min (µIU/mL)	30.67 ± 16.25	11.49 ± 5.19	<0.001
Insulin—OGTT 60 min (µIU/mL)	174.29 ± 93.12	72.32 ± 53.52	0.002
Insulin—OGTT 120 min (µIU/mL)	159.91 ± 83.56	52.41 ± 26.94	0.001
HOMA-IR	6.23 ± 3.25	2.43 ± 1.19	0.001
Insulin resistance, n (%)	13(92.86%)	4(40%)	0.009
Free fatty acids—OGTT 0 min (mmol/L)	0.93 ± 0.18	0.75 ± 0.2	0.159
Free fatty acids—OGTT 60 min (mmol/L)	0.41 ± 0.15	0.32 ± 0.08	0.093
Free fatty acids—OGTT 120 min (mmol/L)	0.34 ± 0.12	0.25 ± 0.1	0.103
Mean systolic blood pressure (mmHg)	118 ± 9.23	108.9 ± 5.43	0.006
Mean systolic blood pressure (percentile)	73.14 ± 8.195	44.7 ± 16.97	0.001
Mean diastolic blood pressure (mmHg)	66.64 ± 7.7	63.5 ± 3.89	0.205
Mean diastolic blood pressure (percentile)	53.86 ± 21.96	41.8 ± 17.03	0.145
Mean heart rate	78.29 ± 8.68	73.9 ± 5.3	0.14

**Table 3 nutrients-13-01683-t003:** Correlation results of anthropometric parameters with *FTO* gene methylation, expression, and *PLAG1* gene expression.

	*FTO* Gene Region 1 Methylation		*FTO* Expression		*PLAG1* Expression	
Anthropometric Parameters	Spearman’s Correlation Coefficient r	*p* Value	Spearman’s Correlation Coefficient r	*p* Value	Spearman’s Correlation Coefficient r	*p* Value
BMI (percentile)	0.428	0.029	0.449	0.021	0.363	0.068
BMI (kg/m^2^)	0.376	0.059	0.608	0.001	0.445	0.023
BMI SD	0.415	0.035	0.473	0.015	0.430	0.028
Waist circumference (cm)	0.370	0.062	0.445	0.023	0.267	0.187
Waist circumference (percentile)	0.504	0.014	0.509	0.013	0.369	0.083
BF_kg	0.437	0.061	0.704	<0.001	0.246	0.311
BF_%	0.289	0.229	0.837	<0.001	0.523	0.022

**Table 4 nutrients-13-01683-t004:** Correlation results of biochemical parameters with *FTO* gene methylation, expression, and *PLAG1* gene expression.

	*FTO* Gene Region 1 Methylation		*FTO* Expression		*PLAG1* Expression	
Biochemical Parameters	Spearman’s Correlation Coefficient r	*p* Value	Spearman’s Correlation Coefficient r	*p* Value	Spearman’s Correlation Coefficient r	*p* Value
Triglycerides (mmol/L)	0.474	0.019	0.314	0.135	0.352	0.092
Free fatty acids—OGTT 0 min (mmol/L)	−0.094	0.719	0.661	0.004	0.563	0.019
Glucose—OGTT 0 min (mmol/L)	−0.320	0.127	−0.566	0.004	−0.002	0.994
Insulin—OGTT 0 min (µIU/mL)	0.428	0.037	0.566	0.004	0.348	0.1
Insulin—OGTT 60 min (µIU/mL)	0.264	0.213	0.444	0.03	0.177	0.408
Insulin—OGTT 120 min (µIU/mL)	0.269	0.215	0.458	0.028	0.221	0.311
HOMA-IR	0.344	0.1	0.442	0.03	0.322	0.125

**Table 5 nutrients-13-01683-t005:** Effect of *FTO* gene region 1 methylation, *FTO* expression, and *PLAG1* expression on the presence of insulin resistance. All calculated *p* values were adjusted for age and sex of the children in ANCOVA.

		*FTO* Gene Region 1 Methylation		*FTO* Expression		*PLAG1* Expression	
		Mean ± SD	*p*_adj_ Value	Mean ± SD	*p_a_*_dj_ Value	Mean ± SD	*p*_adj_ Value
Insulin resistance	Present, *n* = 17	4.79 ± 4.38	0.348	302.63 ± 78.95	0.037	87.68 ± 24.04	0.125
	Absent, *n* = 7	3.62 ± 2.91		208.41 ± 64.84		70.81 ± 12.62	

**Table 6 nutrients-13-01683-t006:** Multiple logistic regression for obesity and insulin resistance depending on genes expression or *FTO* methylation. In each model the main variable was adjusted for age and sex of children, and for any interaction existing between the covariates.

	*FTO* Gene Region 1 Methylation		*FTO* Expression		*PLAG1* Expression	
	OR (95%CI)	*p* Value	OR (95%CI)	*p* Value	OR (95%CI)	*p* Value
Obesity	1.52 (0.97–2.39)	0.066	4.33 (1.38–13.64)	0.012	-	0.129
Insulin resistance	0.041 (0.01–7.26)	0.227	2.33 (1.01–5.40)	0.048	2.67 (0.67–10.62)	0.165

The OR value was calculated for raw value of *FTO* gene region 1 methylation. For *FTO* expression calculations were conducted per increment of 50 units, while for *PLAG1* expression the increment was 10 units.

**Table 7 nutrients-13-01683-t007:** Multiple regression adjusting genes expression and *FTO* region 1 methylation for age and sex of children. If age or sex significantly explained the given model, then standardized regression coefficient and *p*-value for them are shown here.

	*FTO* Region 1 Methylation			*FTO* Expression			*PLAG1* Expression		
	Standardized Regression Coefficient ± SEM, *p*-Value)	R^2^_adj_	*p*-Value for R^2^	Standardized Regression Coefficient ± SEM, *p*-Value)	R^2^_adj_	*p*-Value for R^2^	Standardized Regression Coefficient ± SEM, *p*-Value)	R^2^_adj_	*p*-Value for R^2^
BMI (kg/m^2^)	0.32 ± 0.20, 0.117	0.057	0.242	0.68 ± 0.17, <0.001	0.386	0.003	0.46 ± 0.19, 0.021	0.174	0.067
BMI percentile	0.34 ± 0.20, 0.113	<0.01	0.411	0.72 ± 0.17, <0.001 *Sex*: −0.35 ± 0.17, 0.049	0.368	0.004	0.47 ± 0.19, 0.025	0.112	0.137
BMI SD	0.47 ± 0.20, 0.031	0.11	0.156	0.79 ± 0.17, <0.001	0.445	0.002	0.37 ± 0.22, 0.103	0.013	0.371
Waist circumference (cm)	0.44 ± 0.17, 0.022	0.230	0.032	0.46 ± 0.19, 0.026	0.21	0.039	0.25 ± 0.20, 0.212	0.082	0.187
Waist circumference percentile	0.45 ± 0.21, 0.045	0.073	0.227	0.65 ± 0.20, 0.005	0.25	0.038	0.36 ± 0.22, 0.120	>−0.01	0.444
BF_kg	0.46 ± 0.20, 0.032 *Age*: 0.47 ± 0.18, 0.022	0.423	0.01	0.75 ± 0.11, <0.001; *Age*: 0.42 ± 0.11, 0.001	0.80	<0.001	0.36 ± 0.20, 0.101 *Age:* 0.55 ± 0.20, 0.013	0.325	0.031
BF_%	0.21 ± 0.22, 0.354 *Sex*: 0.49 ± 0.22, 0.041	0.257	0.060	0.76 ± 0.12, <0.001 *Age*: 0.29 ± 0.11, 0.021	0.785	<0.001	0.54 ± 0.17, 0.006 *Age:* 0.42 ± 0.16, 0.022	0.531	0.002
Triglycerides (mmol/L)	0.41 ± 0.22, 0.073	0.031	0.321	0.37 ± 0.22, 0.107	>−0.01	0.415	0.35 ± 0.22, 0.115	>−0.01	0.434
Free fatty acids—OGTT 0 min (mmol/L)	−0.11 ± 0.26, 0.677	<0.01	0.418	0.63 ± 0.23, 0.016	0.386	0.025	0.50 ± 0.21, 0.033	0.311	0.050
Glucose—OGTT 0 min (mmol/L)	−0.23 ± 0.23, 0.299	−0.06	0.648	−0.41 ± 0.22, 0.080	0.037	0.305	0.10 ± 0.23, 0.646	−0.123	0.920
Insulin—OGTT 0 min (μIU/mL)	0.37 ± 0.21, 0.101	0.053	0.264	0.52 ± 0.20, 0.016	0.191	0.065	0.09 ± 0.22, 0.700	−0.088	0.768
Insulin—OGTT 60 min (μIU/mL)	0.30 ± 0.22, 0.20	−0.03	0.530	0.48 ± 0.21, 0.034	0.106	0.161	−0.22 ± 0.21, 0.314	>−0.01	0.428
Insulin—OGTT 120 min (μIU/mL)	0.30 ± 0.23, 0.195	−0.03	0.52	0.43 ± 0.22, 0.068	0.056	0.264	0.08 ± 0.23, 0.747	−0.123	0.898
HOMA-IR	0.30 ± 0.22, 0.184	>−0.01	0.429	0.42 ± 0.22, 0.069	0.072	0.222	0.18 ± 0.22, 0.427	−0.065	0.664

## Data Availability

The datasets generated for this study are available on request to the corresponding author.

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
