# Peer review of "Methylation and Expression of FTO and PLAG1 Genes in Childhood Obesity: Insight into Anthropometric Parameters and Glucose–Lipid Metabolism"

_nutrients, 2021, doi:10.3390/nu13051683_

Round 1

Reviewer 1 Report

The authors performed a cross-sectional analysis of the DNA methylation (DNAm) of 2 genes and pediatric obesity phenotype; and DNAm of FTO and both gene expression on several obesity-related biomarkers.

Major comments:

  1. The biggest concern is the small sample size, which largely limit the validity of the results. The plots in Figure 1-3 also confirms the concern. The distribution of the plotted dots are quite sparsely distributed, the lines are poorly fitted.
  2. More background needs to be mentioned for the PLAG1 gene. Why selects this gene?
  3. Why just analyzed the methylation of FTO, while both gene expression for FTO and PLAG1? This needs to be clearly stated.

Other minor comments:

  1. The fonts in the Tables are not consistent.
  2. Some of the decimal points were presented as “.”, while some were “,”.

Author Response

Thank you for your comments and suggestions.

Point 1. The biggest concern is the small sample size, which largely limit the validity of the results. The plots in Figure 1-3 also confirms the concern. The distribution of the plotted dots are quite sparsely distributed, the lines are poorly fitted.

Response 1. Unfortunately we had not sufficient funds to perform the analysis in larger group of patients. We agree that it limits the validity of the study. We have mentioned it in “Limitations”. The figures were corrected.

Point 2. More background needs to be mentioned for the PLAG1 gene. Why selects this gene?

Response 2: Initially we have chosen the PLAG1 gene as a control to FTO analysis results. As first reports concerning the role of  PLAG1 gene in pathogenesis of obesity were published, we have performed additional analysis of PLAG1. The role of PLAG1 gene is mentioned in introduction and discussion.

Point 3. Why just analyzed the methylation of FTO, while both gene expression for FTO and PLAG1? This needs to be clearly stated.

Response 3: We determined the chosen fragments of both FTO and PLAG1 gene methylation. Due to the lack of a statistically significant difference in the mean and median methylation percentage for PLAG1 region 1 in obese and control groups, Spearman’s correlations with studied parameters were not calculated (line 230-233). The correlation between the methylation level of the tested PLAG1 gene fragment and the level of its expression was not found, which suggests the lack of influence of this gene fragment methylation on the amount of the synthesized mRNA (line 244-246). We did not perform correlation analysis of methylation of the PLAG1 gene with anthropometric measurements, parameters of glucose lipid metabolism, and blood pressure values, for the reason of the found very low methylation degree of the studied PLAG1 region (line 518-521).

Other minor comments:

  1. The fonts in the Tables are not consistent.
  2. Some of the decimal points were presented as “.”, while some were “,”.

Response: Punctuation corrections and other minor language improvements were added.

Reviewer 2 Report

The study is original and is well designed and presented. The only limitation is the small number of patients.

Here I give some suggestions to improve the paper:

Line 48: I think dozen percent is not correct. It shold be replaced by 12%.

You should state the obesity diagnostic criteria used.

Line 119: to replace l by L in 30 microl.

Line 165: to replace ml by mL and mmol/l by mmol/L

I think you can work out BMI in SDS and see if the correlations change.

Line 261: To change were statistically insignificant by were not statistically significant.

Line 317: To change that by this.

Line 375: To change reviled study by study revealed

In general English writting needs to improve

Author Response

Thank you for your comments and suggestions.

The list of corrections is attached.

Round 2

Reviewer 1 Report

The authors provided sufficient replies and the manuscript has been improved. I have no further comments. 

Author Response

Thank you for the response